# Our Life Is Not Here: Migration and Return of Young Spaniards Living in Chile

**Rubén Rodríguez Puertas and Alexandra Ainz Galende \***

Department of Geography, History and Humanities, University of Almeria, La Cañada de San Urbano, 04120 Almeria, Spain; rubenrp@ual.es
\* Correspondence: aag486@ual.es

**Abstract:** With the aim of understanding the recent migration processes of young Spaniards settled in Chile, the present paper analyzes, on the one hand, how these young people experience their arrival and establishment in said Latin American country and, in the other hand, how the process of returning and readjusting to Spanish society takes place. For that, and following the procedures of the Grounded Theory, the discourses of 37 Spanish migrants obtained through in depth interviews were analyzed: 22 of them are living in Chile and the other 15 returned to Spain after spending a long period in Chilean society and have been living in Spain for at least one year since then. All of them have university degrees, are between 25 and 35 years old, and arrived in Chile between 2013 and 2018. This qualitative study shows the way in which these migrants experience their sociocultural integration in Chilean society, which could be typified as "nostalgic" since it is characterized by the idealization of and the longing for their society of origin. Another key characteristic is the eventual return to the country of origin, in which the desynchronization they experience is especially remarkable: after a long period abroad, they feel disconnected from the transformations that have taken place in their original environment, which leads them to experience a difficult process of readjustment to Spanish society that sometimes is even more complex than that experienced abroad.

**Keywords:** acculturation; adaptation; Chile; migrant; Spanish migration

## 1. Introduction

Since the outbreak of the global economic crisis in 2008, Spain began to suffer a significant deterioration in various aspects that affected the social reality of young Spaniards, especially work environment. This situation worsened their already precarious life conditions and drove them to a scenario of chronic unemployment and instability. That deterioration of young people's social conditions and quality of life has transformed not only their social patterns—impossibility of emancipating from parents, difficulty to build their professional identity, showing stress or anxiety, etc.—but also their behavior and perception of the world, with the emergence of a migration discourse that affects especially those with greater education level (Aragón and Bretones 2020; González-Leonardo 2020). In recent years, this conduct has been reflected in the strong increase in Spanish emigration in general and youth emigration in particular, having the emigration of young people between 25 and 34 years of age reached significant numbers during the period 2010–2015. Thus, as shown in Figure 1, even though those numbers decreased in 2016 and 2017, 2018 shows again an increase, being this a trend that will predictably continue due to the present situation caused by the COVID-19 pandemic: the reach of this disease is global, but its consequences in Spanish labor market are catastrophic since this country bases much of its economy on tourism and the hospitality industry, sectors both with great difficulties to adapt to teleworking.

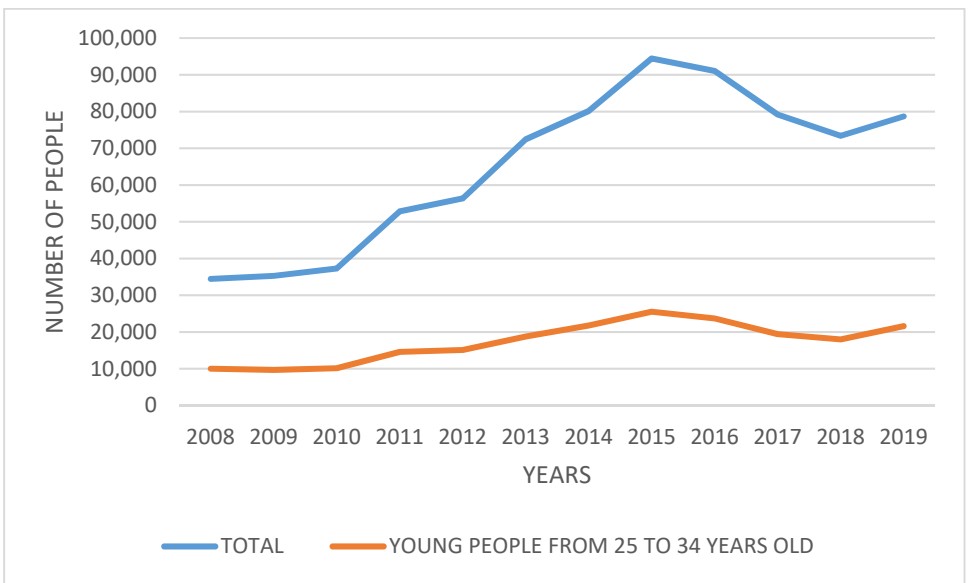

**Figure 1.** Spanish emigration and Spanish youth emigration (25–34 years old) during the period 2008–2019. Source: compiled by the authors based on data from the Spanish National Statistics Institute (Instituto Nacional de Estadística 2019).

This hypothesis is based on the report "Juventud en riesgo: análisis de las consecuencias socioeconómicas de la COVID-19 sobre la población joven en España" [Young people at risk: analysis of the socioeconomic consequences of COVID-19 on the young population in Spain] (Instituto de la Juventud y Consejo de la Juventud de España 2020). This report shows how young Spaniards are the ones who suffer the most from the economic effects of the pandemic, mainly because it affects the tourism and hospitality sector, an oversized sector in Spain—given that it represents 12% of the Gross Domestic Product, one of the highest percentages in the European Community—in which 33.3% of young Spaniards were working in 2020. Therefore, although we do not have data on Spanish emigration in the last year, the authors of this report predict that this crisis may encourage the departure of young Spaniards to other countries whose economy has been less affected by this pandemic.

Thus, due to this difficult context, from 2010 onwards thousands of young people began to leave Spain seeking to overcome the instability and lack of professional development to which they were doomed there, where they adapted their life projects to insecure jobs and ways of life, being unable to achieve a consolidated professional identity that would allow them to consider establishing a life project of their own and emancipating from their parents (Angulo 2020; Bessant et al. 2018; Standing 2014).

That way, although in the most critical years of the economic recession this new Spanish emigrants moved mainly to other European countries, it is also striking how emigration rates reached significant figures considering Latin America as destination (Pérez-Caramés et al. 2018). Thus, as shown in Figure 2, more than 30,000 young Spaniards emigrated to that region between 2010 and 2017, being 2014 and 2015 the years in which more emigrations were recorded with a total of 11,048. These data show how the economic boom that many Latin American countries—mainly Peru, Colombia and Chile—were experiencing during those years, along with the cultural proximity that makes the language barrier not to be an obstacle, led to this increase in the migratory flows of young Spaniards to Latin America.

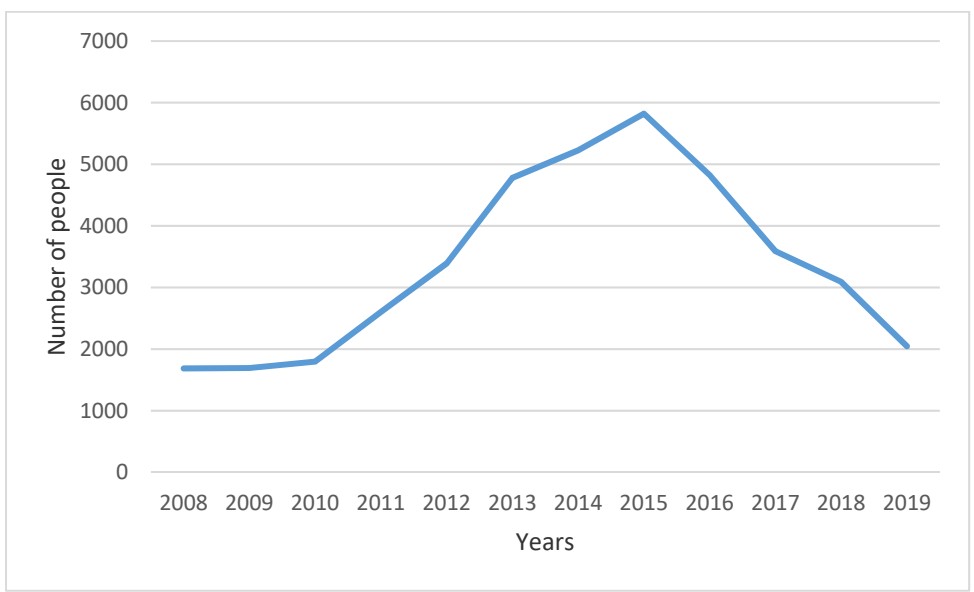

**Figure 2.** Emigration of Spaniards aged 25 to 34 to Latin America. Source: compiled by the authors based on data from the Spanish National Statistics Institute (Instituto Nacional de Estadística 2019).

This trend is explained by the study Migratory Routes and Dynamics between Latin American and Caribbean (LAC) Countries and between LAC and the European Union (Córdova 2015), carried out by the International Organization for Migration (IOM), showing that most of the migration to Latin America comes from Spain. According to this report, just over 7000 Spaniards emigrated to Latin America in 2003, a figure that increased to 154,000 people in 2012. According to this study, the profile of those who emigrate to LAC does not correspond to that of Latin Americans who have acquired some European nationality and decide to return, but rather to that of native and qualified Europeans escaping from unemployment and job insecurity. Thus, it can be affirmed that the economic crisis has caused an important change in the global migration pattern, so that an increase in the Europe-Latin America flow can be observed, as opposed to a decrease in Latin American migrations to Europe (Sassone and Yépez 2014). This is also partly due to the increase in the number of subsidiaries—or satellite companies—of European corporations in Latin America, which, being delocalized in this territory, have not suffered as strongly from the effects of the crisis as their European counterparts and have attracted qualified people from Europe.

The Spanish National Statistics Institute (INE, in Spanish), through the Residential Variation Statistics (EVR, in Spanish), tells us that between 2008 and 2015 there were 196,197 Spaniards that emigrated to the American continent (Instituto Nacional de Estadística 2019). Among these emigrations, especially noteworthy are those of young people to Chile—ranking third in 2014 as a destination for Spaniards aged 25 to 34, after the United States and Ecuador. These young Spaniards were mainly attracted by Chile's economic growth, its infrastructure development, and by the fact that Chile and Spain share a common language. Thus, according to the Department of Foreigners and Migration of Chile (Instituto Nacional de Estadísdtica de Chile 2013), in 2013 Spain was established as the fifth country of origin of immigrant population (3.7%), after Peru (29.7%), Bolivia (20.3%), Colombia (20.2%) and Argentina (4.5%).

This increase in Spanish emigration to Chile also finds explanation in its relatively low unemployment rate: 6.1% in 2013 according to the Chilean National Institute of Statistics, while in Spain that rate was 26.9% for that same year (Instituto Nacional de Estadística 2013). This is mainly due to the strong expansionary cycle of the mining sector, which has been consolidated as the main driving force of the Chilean economy, allowing the arrival of professionals specialized in mining, geology, energy and new technologies. That way, the

increase in Spanish emigration since 2010 is largely due to the implementation of policies to attract skilled labor by the Chilean government with the aim of boosting the growth cycle of said sector. This increase in the number of Spanish emigrants reached its peak in 2013, when various Spanish media published that Chile would offer thousands of specialized job positions between 2013 and 2020 (Rodríguez-Fariñas et al. 2015).

This youth emigration can be included in the so called North-South migration, i.e., young, qualified people that emigrate to the so called emerging economies of southern countries searching for job opportunities they cannot find in their countries of origin due to the strong economic crisis (Masanet and Moncusí 2020). Likewise, another possible explanation can be found in the 'world-systems theory', in which authors such as Saskia Sassen (2007) show that the expansion of the economic globalization has made enterprises to spread their operations over several countries. That way, while in the past the main location for these operations was the head office, there are now other locations: those occupied by the companies hired by the head office to carry out the most complex and/or specialized functions. This organization of the global economy generates a strong interdependence between all countries, with a sharp increase in cross-border transactions and the mobility of labor, resources and knowledge. Thus, several Spanish companies in the construction, telecommunications, mining and energy sectors have subsidiaries in Chile and in several Latin American countries (Sánchez 2016), which facilitated the expatriation of qualified Spanish workers to these countries or their decision to emigrate when those Spanish companies began to be affected by the economic crisis while their subsidiaries abroad remained stable.

It is worth noting that the present research on recent Spanish migration to Chile is a relevant study of great interest in the field of social sciences. This is so because there is scarce literature on the current migration flows between Europe and Latin America (Sassone and Yépez 2014; Córdova 2015; Pérez-Caramés et al. 2018; Rodríguez-Fariñas 2018), and said literature focuses mainly on migration factors and labor market insertion, without dealing in depth, as our study does, with the adaptation processes in the host society and the return process, which makes this research and its results more complex and dynamic, since it contemplates all stages of the migration process—first steps, adaptation process and return.

## 2. Framework of Analysis: Convert, Nostalgic or Cosmopolitan: Profiles of the New Spanish Emigration

There is a recent empirical model which is very useful for understanding the transformations that the perceptions and subjectivities of new Spanish emigrants undergo in the course of their migration processes. This model, which we have used in order to analyze how the young protagonists of this new Spanish emigration experience their sociocultural integration in other countries, has been proposed by Rubén Rodríguez-Puertas (Rodríguez-Puertas 2019; Entrena-Durán and Rodríguez-Puertas 2016), and it differentiates three types of migrants based on their adaptation: converts, nostalgics and cosmopolitans. Thus, according to these typologies, the perception of these young migrants moves between what we call convert adaptation—emphasis on the present or the here-and-now—nostalgic adaptation—idealization of the past or the there-and-then—and the arrival—or not—at a representation of reality halfway between the past lived and the present being lived—cosmopolitan adaptation.

Thus, we can say that "convert migrants" are those who, during the process of adaptation to the new society, usually experience a notable improvement in their socioeconomic status, which leads them to idealize the new environment in which they find themselves for having allowed them to advance in their professional career and improve their social conditions—thus emphasizing the here-and-now—while at the same time they feel a certain phobia towards their country of origin for not having allowed them to develop a life project close to their family and friends—feeling anger for the weakening of the basic social ties they had in Spain. This is a behavior that many sociologists call "nouveau riche syndrome", i.e., on those occasions when migrants achieve in the host country a social

status in accordance with their educational level, they experience a change in values and perceptions: they feel they belong to a new social class and identify themselves with the native—and wealthy—population of the new country.

Sometimes, this "reconversion" can lead migrants to distance themselves from their compatriots also settled in the new society. This happens mainly because, since most of the latter have precarious jobs, *nouveau riche* migrants perceive them as workers belonging to a lower social status.

On the other hand, "nostalgic migrants" would be those who, after being affected by the adaptive stress generated by the various migratory barriers present in the new environment—unfamiliarity with the language, difficulty in getting used to the climate, cultural shock, feeling discriminated, etc.—tend to take refuge in a peer group consisting of other Spanish emigrants. Thus, they recreate a parallel context in which they constantly idealize Spanish society—the there-and-then where those barriers or adversities did not exist. This is so because in this parallel context—which coexists with other contexts of the new society—migrants tend to idealize their country of origin, longing for the comfort environments that they have lost after emigrating: the family and the primary social ties that they had in Spain.

Lastly, "cosmopolitan migrants" are those who, after having spent a long period abroad, suffer a strong maladjustment to their original environment. This is a process of acculturation that leads them to perceive themselves between two realities: on the one hand, they long for their country of origin but do not find their place in it due to the value and cultural transformations they have experienced, and on the other hand, in their resocialization processes they have become accustomed to social relations with people from other countries who share similar situations—other young migrants of cosmopolitan character. This promotes the development of a cultural empathy of a global nature that favors this type of migrant to show greater comfort in integrating into new societies, thus being more difficult for them to acquire strong feelings of nostalgia.

In the following table (see Table 1) we can see the characteristics of each type of adaptation and the subjectivity experienced in each of them, as well as the codes that help to explain them.

**Table 1.** Different types of subjectivities and adaptation processes.

| Explanatory Code | Subjectivity Experienced | Adaptation Process |
|---|---|---|
| Social unease in country of origin (frustration, pessimism) | **Nativophobia** Category that explains the mechanisms that lead to the emergence of a disposition to emigrate. It refers to the Spanish context, marked by unemployment and precariousness, which generates pessimism and frustration in young graduates due to the difficulty of finding a job in accordance with their qualifications and a stable life project. | **Convert Adaptation** Adaptation process in which migrants experience an improvement in their socioeconomic status. This leads them to idealize the new society—for giving them the opportunity to develop a stable life project and an improvement in his social position—and to blame the society of origin—for the loss of his basic primary relationships, constituted by family and friends. Thus, they reconvert their perceptions, identifying themselves with the new society and distancing themselves from the society of origin. |
| Revulsion at/abhorrence of the society of origin | | |
| Positive image of leaving Spain (migratory suggestion) | | |
| Returning equals failure | | |
| Need for family emancipation | | |
| Awful working conditions in country of origin (insecurity, unemployment) | | |
| Identification with the host population | | |
| Convert/*nouveau riche* syndrome | | |
| Equitable place to raise a family and educate children | | |

**Table 1.** *Cont.*

| Explanatory Code | Subjectivity Experienced | Adaptation Process |
|---|---|---|
| Migratory barriers | **Nativophilia**<br>Category that explains how the initial clash with the host society takes place once the migrants arrive in it. Young migrants feel that they are between two realities. Thus, although they are not physically in their country of origin, they remain mentally rooted to their homeland. Therefore, they try to seek an immediate environment similar to the one they left behind. | **Nostalgic Adaptation**<br>Adaptation process in which migrants, affected by the adaptive stress caused by the difficulty of overcoming the multiple barriers inherent to the first steps in a new society, tend to take refuge in a comfort group consisting of other migrant Spaniards. This creates a context in which Spanish society tends to be idealized, so migrants attempt to reproduce its patterns. They experience some kind of attempt to reterritorialize said patterns, while at the same time their desire to return is increasingly intensified. |
| Language barriers | | |
| Feeling discriminated | | |
| Taking refuge in peer group consisting of other Spaniards | | |
| Attempts of reterritorializing social patterns/customs of origin | | |
| Unfavorable climate | | |
| Idealization of life in country of origin | | |
| Idealization of the return | | |
| Identification with the population of origin | | |
| Perception of 'no place' to raise a family and educate children | | |
| Maladjustment to culture and society of origin | **Hybrid Subjectivity**<br>This category explains the transformations that take place in the migrants' subjectivity after a long period in the host society. A process of acculturation/maladjustment to the country of origin begins. Migrants experience an internal conflict that can lead them to a reinforcement of the nativophobia or to the mutation to a new, hybrid subjectivity. | **Cosmopolitan Adaptation**<br>Process in which migrants, due to their continuous interaction with people from different countries and several social spaces, assume an attitude and cultural empathy of a cosmopolitan nature and, therefore, of a more or less global scope. This makes it easier for them to integrate into the new society, thus weakening their feelings of nostalgia. |
| Multiple identity (various social spaces of identification) | | |
| Deterritorialized identity | | |
| Cosmopolitanism | | |
| Estrangement and lack of referents to hold on to | | |
| Short-term life projects | | |
| Cultural empathy/global perception | | |

The aim of this paper is to know how young Spaniards who arrived in Chile between 2013 and 2018 face and experience their migration processes—adaptation in Chilean society and readjustment to Spanish society—as well as trying to identify what role the aforementioned typologies of adaptation play in said migration processes.

However, as we have already mentioned, in addition to the adaptation process in the host society, this study analyzes how the return and readaptation to the society of origin take place. According to the literature about the return process, in general, there are three different patterns: people who return after a short stay, those who return after fulfilling their migration objectives, and those who decide to settle permanently in the host country (Elgorriaga et al. 2020; Josa-Prado and Gómez-Sebastián 2019).

Thus, the big question would be the following: What are the factors that predispose migrants to return to their country of origin? In this case, we find several conditioning factors: failing to fulfill migration expectations; precarious working conditions; possibility of finding a job in the country of origin; distancing from the family; contacts and ties with the country of origin; and the lack of social, political or cultural ties with the host country.

In the Spanish context, and according to the report on the return and incorporation of scientists and researchers in Spain (Josa-Prado and Gómez-Sebastián 2019), there is a great amount of qualified young people currently working abroad, and they present three main typologies: those who are in a training period—master's degree, doctoral or postdoctoral students—those who have made the decision to develop their career abroad; and those who return but, in most cases, decide to leave again. According to that report, the key to both the taking the decision to emigrate and the experiencing difficulties after returning lies in the combination of the high unemployment rate that the sectors with the greatest

number of qualified people suffer in Spain, together with non-competitive salaries, having a status below what would correspond to the professional qualification obtained, and the lack of opportunities for professional development (Arango 2016; Domínguez-Mujica and Díaz-Hernández 2019).

As can be seen, much of the literature on return migration focuses mainly on understanding the factors that led migrants to leave, which conditions must be met for them to return—or not—and how labor market insertion takes place in the society of origin. In line with these ideas, the present research also focuses on these factors but, above all, on analyzing the reverse cultural shock that happens during the readaptation to the society of origin, which sometimes generates a significant loss of identity in the migrant.

## 3. Materials and Methods

With the intention of understanding how the migration and adaptation processes of young Spaniards living in the Chilean society and the readjustment processes to the Spanish society of those who decided to return are like, we conducted 22 in-depth interviews to Spaniards between 25 and 35 years old who had an university degree and were living in Chile, where they arrived between 2013 and 2018, as well as 15 in-depth interviews to young Spaniards with the same characteristics that had returned to Spain more than a year prior to the interviews after having spent a long period living in Chile (see Tables 2 and 3).

**Table 2.** Profiles of the interviewees living in Chile.

| Age | Gender | Time Living in Chile (Months) | Professional Education | Profession |
| --- | --- | --- | --- | --- |
| 33 | Male | 48 | Degree in Economics | Consultant at new technologies company |
| 27 | Female | 36 | Degree in Teaching | Teacher at municipal school |
| 31 | Female | 44 | Degree in Labor Relations | Human resources manager |
| 28 | Female | 42 | Degree in Geology | Manager at mining company (Iquique, northern Chile) |
| 31 | Female | 39 | Degree in Environmental Science | Environmental project technician |
| 29 | Male | 41 | Architecture | Construction site manager |
| 31 | Male | 38 | Degree in Social Work | Community development technician |
| 28 | Male | 42 | Technical Architecture | University professor |
| 27 | Male | 38 | Degree in Electrical Engineering | Electrical project engineer |
| 29 | Male | 40 | Degree in Geology | Hydrogeologist at mining company |
| 30 | Female | 38 | Degree in Pharmacy | Pharmacist |
| 28 | Male | 38 | Degree in Geology | Field hydrogeologist |
| 31 | Female | 42 | Degree in Chemistry | University researcher |
| 31 | Female | 45 | Degree in IT | Cybersecurity technician |

**Table 2.** *Cont.*

| Age | Gender | Time Living in Chile (Months) | Professional Education | Profession |
|---|---|---|---|---|
| 32 | Male | 39 | Degree in Economics | Export technician |
| 33 | Male | 40 | Technical Architecture | Office manager at engineering company |
| 29 | Female | 38 | Degree in Sociology | Postdoctoral researcher |
| 30 | Male | 39 | Forest Engineering | Environmental consultant |
| 32 | Male | 38 | Degree in Chemistry | Logistics manager at chemical company |
| 31 | Female | 38 | Degree in Geology | Mining site manager (Copiapó, Chile) |
| 30 | Male | 39 | Degree in Environmental Science | Environmental engineer |
| 31 | Male | 37 | Degree in Mathematics | Mathematics teacher |

**Table 3.** Profiles of the interviewees that returned to Spain.

| Age | Gender | Time Spent in Chile (Months) | Professional Education | Time Living in Spain (Months) |
|---|---|---|---|---|
| 33 | Male | 40 | Degree in Geology | 18 |
| 32 | Female | 38 | Degree in Teaching | 30 |
| 31 | Male | 44 | Architecture | 24 |
| 30 | Female | 39 | Degree in Geology | 15 |
| 33 | Female | 40 | Degree in Social Work | 19 |
| 31 | Female | 41 | Technical Architecture | 22 |
| 31 | Male | 39 | Degree in Economics | 20 |
| 32 | Male | 40 | Technical Architecture | 19 |
| 33 | Male | 40 | Degree in Geology | 22 |
| 29 | Male | 39 | Degree in Social Work | 14 |
| 32 | Male | 39 | Degree in Social Work | 15 |
| 29 | Female | 39 | Degree in Psychology | 16 |
| 31 | Female | 42 | Degree in Geology | 19 |
| 32 | Female | 40 | Technical Architecture | 22 |
| 32 | Female | 44 | Degree in Social Work | 24 |

The 25 to 35 age bracket was selected because, according to the Residential Variation Statistics from the Spanish National Statistics Institute (Instituto Nacional de Estadística 2019), this interval is the one with the highest percentage of emigrations (around 30%) during the period studied (2013–2018). Likewise, it was decided that they should be young people with university degrees because various opinion barometers in Spain (Centro de Investigaciones Sociológicas 2012; Real Instituto Elcano 2013) showed that, during the

hardest years of the economic recession, the young Spaniards most predisposed to emigrate were those with university degrees.

### 3.1. Research Methodology

The research technique used to produce the data analyzed in this study was the in-depth interview. As explained in the previous section, with the aim of finding out about the migration and adaptation processes of young Spaniards living in Chilean society as well as the process of readjustment to Spanish society of those who decided to return, a total of 22 Spanish residents in Chile and 15 Spaniards who returned to Spain after spending a long period in the Latin American country and had been residing in their country of origin for at least one year were interviewed.

These interviews lasted approximately 70 min each. They were based on a script whose objective was to collect life stories, on the one hand, about the way in which young migrants experienced their own migration processes in Chilean society, and on the other hand, how those who decided to return to Spain experienced their readjustment processes in their society of origin. Said script followed a series of approaches that can be classified into the following topics: sociodemographic profiles, life experiences and sociocultural integration in Chilean society, and the perceptions and subjectivities of those who had returned to Spain.

### 3.2. Procedure for Data Production

Given that the researcher was in Spain, the interviews were conducted via the computer program *Skype*. They were recorded using the software *Call Graph* and transcribed verbatim for later analysis. The participants were contacted through platforms of young Spanish emigrants such as *Marea Granate*—Spanish for "Maroon Tide"—or *Spaniards*—self-help platforms that operate both physically and digitally—and by using the method known as snowball sampling, that is, asking informants for help in identifying other people with important traits for the research (Valles 2003; Saunders et al. 2018).

As a methodological novelty, it is important to point out the advantages of using the Skype video-calling program: in addition to the possibility of interviewing participants even if they were in remote spatial contexts, the use of this program made possible a more spontaneous and fluid discourse during the interaction, since the interviewees did not feel as intimidated as may happen during face-to-face communicative processes, which led them to give more information when certain, more sensitive or taboo topics were discussed—mainly when talking about the indigenous population of the host country, discussing the perceived discrimination, or when identity-related feelings arose.

### 3.3. Data Analysis

The analytical categories of the three-dimensional model previously explained in Section 2 were used for the analysis of the data obtained. Said model consists of three fundamental categories: convert adaptation, nostalgic adaptation and cosmopolitan adaptation. This was carried out using Grounded Theory procedures (Glaser and Strauss 1967; Strauss and Corbin 1990), with the intention of ordering the information from the interviewees into the aforementioned precoded categories. Thus, with the use of the software Atlas.Ti6, the discourses produced in the in-depth interviews were classified within each of those categories in order to understand the underlying pattern in the interviewees' discourses. This ultimately led to finding out whether they followed social integration processes marked by rejection of Spanish society—convert adaptation—by strong nostalgia for it—nostalgic adaptation—or if, due to their migratory experiences, they had developed a hybrid identity based on cosmopolitanism or lack of territorial identity—cosmopolitan adaptation.

The interviews followed a script whose objective was to reveal the migrants' perception of their sociocultural integration in Chilean society and their adaptation strategies: rejection of their context of origin for not having been able to develop their life project

there and attachment to Chilean society for giving them the opportunity to develop professionally; nostalgia for the living space they have left behind, caused by the difficulties they experience in their new environment; or cosmopolitan integration and acquisition of a mixed identity made up both of aspects from the society of origin and from the host society, which ultimately led them to develop a sense of belonging of a more global nature.

Thus, after differentiating these three categories of sociocultural integration—convert, nostalgic and cosmopolitan—fieldwork began with the intention of probing into them. Theoretical saturation—the point at which new interviews do not provide additional information relevant to the objective being studied—was reached after conducting 22 in-depth interviews. The discourses of those young people who, after spending a long period in Chile, had returned and have been residing in Spain for at least one year, were likewise analyzed, trying in this case to know how the process of returning and readjusting to the society of origin took place. This task yielded the first significant results after conducting 15 in-depth interviews.

The Grounded Theory was considered useful because the aim of the present work is to know and explain how the sociocultural integration of young Spaniards in Chilean society is developed, as well as to know and explain the process of returning and readjusting to Spanish society, but without the bias that starting from previous hypotheses would imply. Moreover, it is the most appropriate methodology for letting migrants to express themselves freely and, based on their discourses and experiences, to reveal in an approximate manner how they construct those processes.

## 4. Results

### 4.1. We Are Not Emigrants, We Are Exiles: Frustration as a Migration Factor

The reasons that led these young people to emigrate to Chile are mainly linked to the context of frustration and pessimism they experienced in Spain, an environment in which they were either unemployed, or in the best of cases had unpaid internship contracts or developed precarious labor trajectories characterized by salaries that did not allow them to start a life project of their own. This dramatic context where young people are not able to acquire a solid and stable work identity through which they can properly integrate themselves into socioeconomic life, ultimately becomes a great motivation to develop a strong migratory discourse and a feeling of rejection towards the Spanish labor context:

> «After finishing my degree in geology and later specializing in hydrogeology through a postgraduate course I took in Madrid, the company where I was working as an intern without any salary offered me a contract, but with a ridiculous salary. They offered me 600 euros for a basic technician position, but the tasks I would actually carry out would be those of a hydrogeologist, so I should be paid considerably more. So, I said to myself: "no, this is a swindle, I have just finished my studies and I'm not going to stay in Madrid with this salary." Later, I found another job in Barcelona, but only for four hours a week and a ridiculous salary, so I decided I would not accept it because I could not rent an apartment with such a low salary. Then I thought: "that's it, to stay in Spain is to lose my future, if I stay I will never be able to develop as geologist." So I started looking for a job all over the world. I realized that I had no other option but to emigrate». (Hydrogeologist, 29 years old)

This is a discourse in which these young people do not perceive themselves as emigrants, but as "economic exiles," arguing that they did not leave with the intention of seeing the world or having a cosmopolitan life, but that they did so because of governments that did not know how to adequately manage the economic crisis:

> «I did not leave Spain to see the world, I am not an adventurer, I consider myself an exile. Today I feel the obligation to be in Chile, not the possibility of choosing whether I am here or not. I do not see the possibility of returning to Spain, because returning would mean working in something for what I did not study,

being at my parents' house and having much more instability than I already have here». (Pharmacist, 30 years old)

«Although I emigrated voluntarily, prior to that I spent two years looking for work and only found insecure contracts that did not allow me to start my own life, so I don't really know if I left because I wanted to, or almost out of necessity and obligation, to try to escape from a country in which politicians manage a crisis by curtailing citizens' rights and offering us absurd salaries. I see it clearly: we, qualified young people who leave, are not emigrants, we are exiles; we don't leave, they throw us out». (Construction site manager, 29 years old)

*4.2. Ease of Adaptation of Spanish Migrants as a Consequence of Chile's Affinity towards Western and European Societies*

Generally, Spanish migrants residing in Chile experience an easy and problem-free sociocultural integration. This responds to the existence of a social, collective imagination in Chilean population characterized by a strong affinity and/or sympathy towards the European and Western societies. It is an image in which people of European origin are better valued in comparison with those who come from closer territories such as Bolivia, Peru or Colombia. This fact is evident in the discourses of Spanish migrants themselves:

«In Chile, if you are Spanish you are valued better than other nationalities. In this society, there are levels of immigrants. To give you a case, my partner is Peruvian, and she has suffered discrimination. However, a Spaniard will always be better considered than an Argentinean, and an Argentinean will always be better considered than a Colombian, and a Colombian better than a Peruvian, and a Peruvian better than a Bolivian, so there are levels, here in Chile they tend to segregate according to your country of origin. It's very simple, you will be considered a good or bad immigrant based on your skin and origin». (Environmental consultant, 30 years old)

«Spaniards and people coming from Europe or the United States, in general, are perceived as beneficial and positive migration for the growth of the country. We are seen as more competent and serious, more educated. This does not happen to immigrants coming from Bolivia, Haiti or Peru». (University researcher, 31 years old)

This greater appreciation of the European can be seen in studies such as *Bitácora Social II: ¿Aceptación o discriminación en Chile?* (Ubilla et al. 2015) [Social Logbook II: Acceptance or discrimination in Chile?]. That study, in which more than 50,000 people—70.9% of the total population—belonging to the 73 urban communes were interviewed, shows the perception that Chileans have of immigrants depending on their origin. On the one hand, those immigrants perceived with a bad or very bad image are, in order of importance: Bolivians (34%), Peruvians (30%) and Colombians (22%). On the other hand, those perceived with a very positive image are: Germans (76%), North Americans (75%) and Spaniards (74%). We paradoxically see how citizens with whom they share a border are frowned upon, while at the same time those who come from Europe or North America are idealized.

This difference in the perceived value given to migrants in Chilean society responds to the strong admiration that a large part of that society shows towards several countries in aspects such as education, economy and lifestyles. In this line, some authors like Aguayo (2011) explain that Chileans consider as superior those countries with high cultural and educational levels, as is the case of the United States and several European countries—Germany, Spain, France, etc. This admiration entails the idealization of certain physical aspects, such as being taller, having light eyes, white skin, etc. These aspects are linked to distant cultures such as Europe or North America, while the physical features of closer countries—such as Peru or Bolivia—tend to be rejected. In short, this attachment to Western culture, which has a discriminatory effect by idealizing cultures such as the European one,

causes Spanish migrants to be better perceived and, consequently, to have greater ease in overcoming the varied migration barriers.

However, although we have seen that Spanish migrants' sociocultural integration in Chilean society is easier since they do not face significant barriers such as being discriminated—on the contrary, they are considered positive migrants for the country—they do experience some difficulties when integrating into the new environment. This responds to the relevant cultural differences in subjects such as enjoying free time, experiencing personal relations and building a peer group:

> «We enjoy our free time in a totally different way: Chileans spend the weekend at home or preparing a barbecue, that's their main way of spending their free time, while Spaniards are more streetwise, we like moving around, not staying in one place. That creates an important division and it's complicated to make friends, besides the fact that they do not allow you to enter their circle of friends, they exclude you». (Logistics manager at a chemical company, 32 years old)

> «Chileans form their group of friends in a very exclusive way, and they do it during university and high school, and they are friendships that they keep for life. If you were not in those spaces or didn't move in such circles, you are left out. It has to do a lot with the culture here, they are very classist in that sense. You have many areas where you are excluded, and it gives you the feeling of being isolated. Any circle of friends you try to form here, it's never the same as the one you had in Spain». (Office manager at an engineering company, 33 years old)

This difficulty in relating to the native population is added to the perception of a progressive loss of social well-being due to aspects such as poor health care, precarious working conditions, strong environmental pollution, and the existence of strong classism and extreme social poverty. These typical effects of a neoliberal state based on competition lead Spanish migrants to develop a segregation acculturation strategy, i.e., in order to defend themselves against those difficulties, they tend to take refuge in a peer group consisting mostly of other Spanish emigrants, creating a space where they reproduce the cultural patterns of Spain. Ultimately, due to the frustrating discourses about the aforementioned lack of social well-being that they perceive in Chile, they end up idealizing the return. We can observe such idealization of the society of departure in the following interview excerpt:

> «We Spaniards are a real ghetto here, we gather among ourselves, we go only to Spanish restaurants. There are many Spanish bars where there are only Spaniards, and they engage in conversation from a perspective of frustration and discouragement, as if being here was a punishment and they urgently needed to return to Spain. When you spend a lot of time in this context, it is logical that you end up returning». (Logistics manager at a chemical company, 32 years old)

*4.3. The Difficult Decision: Professional Growth in Chile or Social Well-Being in Spain*

These young Spaniards talk positively about how they perceive their first steps in the migration process, usually highlighting aspects such as the professional development and labor stability achieved. That way, they remark that the host environment gives them the opportunity to gain a strong professional identity, in opposition to the uncertainty typical of their experience in Spain, marked by instability and the difficulty to solidly advance in their professional projects. However, this apparently attractive first impression ends up being an important conflict of interests as their migration project gets consolidated after a long period—about three years—living in Chile, caused by the professional growth the new country brings them in opposition to the progressive loss of social well-being they begin to suffer. This aspect can be observed in the discourse of some of the interviewees:

> «The first months in Chile were very good, I felt good, seeing that I was growing professionally and could develop as an architect. But as time goes by you collide with the reality of this country, and you understand that it is practically

impossible to settle and spend the rest of your life in Chile, when you see the existing social conditions here. You understand that if, for example, you have an accident at work or you suffer from an important illness, you will have a debt for life. If you suffer from cancer you will get into debt, and if you don't overcome it, that debt will be passed on to your family. When you think about this neoliberal and selfish system, where aspects such as health and education are only available to a few, you say to yourself: "I need to go back to Spain." You probably won't get a decent job, but if you suffer from an illness you won't get into debt. At least, in Spain there's a welfare state that in many situations makes up for all the disadvantages. And let's not forget the issue of retirement. Here, a nurse, a teacher or a lawyer have practically nothing left for retirement, they have to live as best they can, sometimes even setting up street-food kiosks». (Construction site manager, 29 years old)

These aspects, strongly linked to the market neoliberalism on which Chilean society is based, ultimately cause a large number of Spanish professionals who emigrated to that country to return or re-emigrate to another country. Thus, these young people begin to create in their collective imagination an "ideal plane" where they emphasize the idea that Spain is experiencing an economic and labor improvement, and that thanks to their professional experience developed in Chile they will not have many difficulties in finding a job when they return to Spain:

«I think that Spain is improving quite a lot labor-wise in recent years, that's what I perceive in conversations with my Spanish friends and in what I read in the press, that's why I am so optimist, so it won't be long before I go back. With everything I've learned here I hope I won't have any difficulties in getting a good job as an architect in Spain». (Construction site manager, 29 years old)

Likewise, the cultural shock caused by the socialization that these young people have had in a welfare state such as the Spanish one and the experiences developed in a neoliberal market state generates that they return with a greater social conscience in the defense of public services:

«My experience in Chile makes me really know the value of public health and education. My friends who have stayed in Spain do not have this knowledge, they do not know what it is to pay for basic services that should be for everyone. I am clear that I am returning to Spain to fight, to not let our quality of life to be taken away, so that in the future, Spain is not the same as Chile, socially speaking. A couple of days ago a group of Spaniards who are returning to Spain commented: "we arrived in Chile as frightened lambs, but we return to Spain as true lions"». (Human resources manager, 31 years old)

Finally, another remarkable aspect that shows the strong idealization of the return these young migrants have is that some decisions as important as that of becoming parents are seen as a clear incentive to return to Spain instead of to permanently establish the family in Chile, even when in Chilean society they have professional and economic stability. Fundamentally, this is so because they do not want to raise their children in a society in which privatization of public services prevails, with the consequent increase in social inequalities that it entails:

«It has been a great joy to have our son here. We are doing very well financially and professionally, which is why we have decided to become parents here. But, on the other hand, it is a circumstance that has led us to decide to return to Spain. We do not want our son to grow up in such an unfair, classist and unequal society. Besides, we would have to bear the costs of a private and poor quality education. We have already decided to return, even though we are sad to leave such good jobs and return to the uncertainty of life in Spain. We prefer to put social welfare before our economic growth. A dignified life comes before money». (Community development technician, 31 years old)

*4.4. The Return to Spain: An Unsynchronized Readjustment*

After spending a long period in Chilean society, the return process that these young Spanish migrants experience is mainly characterized by the desynchronization with their country of origin. This happens mainly due to the inconsistency between the image they had of their environment when they undertook their migration project and the new image they find upon their return. Before moving to Chile, these young emigrants had in Spain an environment of comfort and security constituted by family protection and the primary social ties with their peer groups in different areas: work, neighborhood, university, family, etc. These relationships are necessary to generate a solid and constant sense of belonging, which was only threatened by the lack of stable employment they experienced in Spain, a lack of professional identity that was one of the main factors that led them to emigrate.

Thus, when these migrants spend a long period in the host society, they begin to become aware of the loss of these necessary relationships, which sometimes leads them to idealize the past and try to reterritorialize it in their new context. That way, they establish peer groups with other compatriots, in which they recreate some aspects of the Spanish culture both discursively and practically:

«Here in Chile I really missed everything I left behind in Spain: my family, my neighborhood, drinking beer with my friends... I was able to cope with all this because thank God I met many Spaniards with whom I kept the same customs we had in Spain: we went out to eat at Spanish restaurants, watched Spanish movies, followed the Spanish national soccer team matches, even celebrated some Spanish festivities such as *Sanfermines*. That day, one Basque guy in the group bought us some beers, and we partied». (Graduate in social work, 29 years old)

This process, which some authors call "glocalization," involves taking the aspects of the original culture to other places, in order not to lose those roots that are so important to maintain a solid and stable identity (Homobono 2019; Appadurai 1995).

The problem faced by these migrants is that, when they arrive in Spain, they realize that the past they had idealized during their stay in Chile is no longer the same as when they left it behind. Thus, they perceive both material changes—physical modifications in their neighborhoods—and relationship changes—weakening of the social ties they had before emigrating—as explained by the returnees themselves:

«When I left I said goodbye to my neighborhood and my friends and, now that I am back after three years, I realize that everything is very different. I was melancholic about coming back, but now that I'm here, I realize that I'm still a foreigner in my own country. My neighborhood is a different place, there's a subway now, the usual stores have closed, they have removed the cinema to build a supermarket... And what has affected me the most, is that not only my neighborhood has changed, but also my friends. The first day I arrived I expected to see them all, but that same day I realized they have evolved, they are very different, many have married and have had children, others have moved ... And of course, it's not like before. Now, meeting and having a couple of beers is mission impossible, they all have other obligations. And when I get to see them they talk about anecdotes that I wasn't able to live, I have become the foreigner, the annoying person who talks about Chile». (Graduate in teaching, 32 years old)

As the previous excerpt shows, migrants experience a lack of synchronization between what they imagine and what they really find upon their return to Spain. This is a disconnection that, as explained above, not only affects the purely material—changes in the environment—but also the transformations that have taken place in the peer group, where all the experiences and changes that this group has undergone during the years when the migrants were absent are non-existent for them. Thus, sometimes, returnees feel an identity void due to this desynchronization:

«When I arrived in Spain I realized that neither my friends nor I were the same. In our conversations, they always remembered moments that I had not lived. I was lost, I did not know what to say, what to think, I was the annoying one who only talked about Chile because those were my experiences. I noticed that I have lost everything they have lived during my absence, it made me feel empty, as if I was no longer from here». (Graduate in geology, 33 years old)

## 5. Conclusions and New Lines of Research

As established in the present paper, these migrants, who call themselves 'labor exiles', seem to follow a sociocultural integration pattern that could be classified in the 'nostalgic adaptation' category, as a result of the loss of social well-being they perceive in Chile. The main factor that make them perceive it is the privatization of services such as health or education, whose costs are individually payed by the citizens; other key factor is the difficulty to establish relationships with native population due primarily to cultural differences in their approach to free time or friendship, what makes them seek refuge in peer groups consisting of other Spanish migrants. That way, they construct personal spaces where, through discourses based on the frustration and pessimism that said loss causes, they idealize their society of origin as well as the eventual return to it.

Although it is true that they show a positive attitude in the beginning of their migration process, when they highlight the work and economic stability they have achieved, once they settle in Chilean society they start to perceive the progressive loss of social well-being and security. In the same way, they start to face some social realities which they find difficult to live with, like for example a strong classism, the high environmental pollution and the health problems it entails, the weak retirement system, the high costs of health and education, the high rates of social unrest, and as a result of all that, the serious situation of extreme poverty that the country suffers. That situation, which young people raised and socialized in a welfare state find difficult to understand, leads them to build a collective imagination in which returning to Spain becomes a priority and a desirable option, even if it entails abandoning the work and economic stability achieved in the host country.

However, as we have seen in this paper, these young migrants experience an unsynchronized readjustment when they return, due to the image of Spain built when they were abroad being different to what they actually find at their arrival in their country of origin, especially with respect to social relationships. Thus, they face relevant changes both in their physical and in their social environments, and especially in their basic, social ties with their peer groups. They experience a feeling of emptiness and lack of identity, because during their absence, their place of origin has mutated in such a way that it is totally different to how they remembered it, and even their primary relationships have been transformed, so they end up feeling like foreigners in their own country.

Therefore, the model created by Rubén Rodríguez-Puertas (2019), which shows the versatility of the adaptation processes of young Spanish migrants and the three adaptation strategies that they may develop—convert, nostalgic or cosmopolitan—seems not to apply to the case of those young Spaniards that have emigrated to Chile. There is no such versatility: most of the interviewees opted for an adaptation strategy where nostalgia for the society of origin prevails. As explained above, this is due to their clash with a society based on market neoliberalism, which in most cases makes them develop an integration strategy marked by an intense subjectivity of nostalgic nature. Said subjectivity leads them to seek refuge from the negative experiences they live and perceive by building peer groups consisting mainly of other Spanish emigrants. In those groups, they create contexts in which they idealize Spanish society as well as the return to it, thus seeing said return as the safest and most feasible option. That way, the application of this three-dimensional model to the Chilean context opens new lines of study of this new Spanish youth migration, with the possible incorporation of other relevant factors such as the social well-being the migrants perceive or the confrontation with social, political and economical systems different to those of the country of origin.

It is also worth noting the political implications that the results of this study may have. Thus, in words of the interviewees themselves, the main factors that led to the departure of these qualified migrants were the precariousness and temporary nature of the Spanish labor market. Therefore, all government actions and policies carried out to curb these departures should prioritize two objectives: fighting precariousness and opting for a competitive model not based on temporary employment and low salaries. In addition to that, with regard to the reverse cultural shock experienced by these young people after their return, this study highlights the need to provide them with psychological support to manage the feelings and emotions derived from the weakening and/or loss of the identities, relationships and social ties they had in Spain, and which have been greatly transformed during this process.

**Author Contributions:** R.R.P. and A.A.G. carried out all the stages of the paper: conceptualization, conception, design, research, analysis and conclusions, writing, and final review. Both authors have read and agreed to the published version of the manuscript.

**Funding:** This research received no external funding.

**Data Availability Statement:** The interviewees stated that they did not want the complete interviews to be share to anyone.

**Acknowledgments:** The authors would like to thank all those who have participated in this project and have agreed to be interviewed, since this research has been possible thanks to their stories and experiences. We would also like to thank the movement *Marea Granate* for providing part of the contacts for the in-depth interviews.

**Conflicts of Interest:** The authors declare no conflict of interest.

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
