# Peer review of "Our Life Is Not Here: Migration and Return of Young Spaniards Living in Chile"

_socsci, doi:10.3390/socsci10080293_

Round 1

Reviewer 1 Report

Overall, the article is well written and has several noteworthy features. Besides these elements, there are some issues to be solved:

  1. What is the link between Figure 1 and COVID-19 pandemics? In Figure 1, the statistics are ending in 2019, while COVID-19 started in 2020! Please comment more on the statement you make: "Thus, as shown in figure 1, even though those numbers decreased in 2016 and 2017, 2018 shows again an increase, being this a trend that will predictably continue due to the present situation caused by the COVID-19 pandemic: the reach of this disease is global, but its consequences in Spanish labor market are catastrophic since this country bases much of its economy on tourism and the hospitality industry, sectors both with great difficulties to adapt to teleworking." Are you sure that, in the pandemics context, the trend is that you are emphasized? Please give more details, statistics and references to support this idea!
  2. Which are the methodological novelties you brought into consideration?
  3. What are the implications of the results of this study?
  4. Do they have any practical / political applicability? If yes, how a policymaker could adopt certain policies to stop the 'exodus' and reintegrate the migrants back home?

Author Response

First of all, the authors would like to thank the reviewer for their considerations and constructive criticism, which have undoubtedly helped to improve the work presented here.

Point 1: What is the link between Figure 1 and COVID-19 pandemics? In Figure 1, the statistics are ending in 2019, while COVID-19 started in 2020! Please comment more on the statement you make: "Thus, as shown in figure 1, even though those numbers decreased in 2016 and 2017, 2018 shows again an increase, being this a trend that will predictably continue due to the present situation caused by the COVID-19 pandemic: the reach of this disease is global, but its consequences in Spanish labor market are catastrophic since this country bases much of its economy on tourism and the hospitality industry, sectors both with great difficulties to adapt to teleworking." Are you sure that, in the pandemics context, the trend is that you are emphasized? Please give more details, statistics and references to support this idea!

Response 1: In response to this suggestion, we have supported the hypothesis that the pandemic may encourage the outflow of young people to other countries, based on data from the report ‘Juventud en riesgo: análisis de las consecuencias de la COVID‑19 sobre la población joven en España’ [Spanish for ‘Youth at Risk: Analysis of the Consequences of COVID‑19 on Young Population in Spain’] by the Spanish institute ‘Instituto de la Juventud’ [Institute of Youth]. In any case, in our article, we have tried to clarify that this is not a statement (we still do not have updated statistics to know if this emigration will really increase), but a hypothesis based on the strong dependence that Spain has on the tourism and hospitality sector (much higher than the other countries of the European Community), being this sector one of the most affected by the current pandemic, and in which a very high percentage of young people work. The newly added reference is in the introduction, just above Figure 1. The report has also been added to the bibliography.

Point 2: Which are the methodological novelties you brought into consideration?

Response 2: In this case, in section 3.2 (procedure for data production), we have mentioned the methodological novelty of using the Skype video‑call program, as well as the main advantages it brings: being able to communicate in real time with people in other countries, and contributing to generate a more spontaneous conversation where the disadvantage of the social‑desirability bias is less prominent (since the researcher is not face to face with the interviewee).

Point 3: What are the implications of the results of this study?

Response 3: The results of this study have implications at three levels:

  • First, they show how part of this emigration responds to a feeling of exile, and therefore, they make visible that one of the main migration factors lies in the precariousness of the Spanish labor market and in the lack of a productive framework capable of incorporating the most qualified labor force.
  • Second, this research shows how the socio‑cultural insertion of these young people in the country of destination takes place, highlighting the importance that migrants give to perceived social welfare (even more so than job stability).
  • And finally, regarding the return process, the discourses of the interviewees show the reverse cultural shock they experience, in which they perceive the loss of their basic social relations and their former spaces of comfort, and therefore, of their identities.

These implications and their political applicability have been added to the section ‘Conclusions and New Lines of Research’.

Point 4: Do they have any practical / political applicability? If yes, how a policymaker could adopt certain policies to stop the 'exodus' and reintegrate the migrants back home?

Response 4: In this case, the research does present some results that can contribute to the application of migration policies:

  • Regarding policies to alleviate this outflow of qualified young people, as the interviewees themselves have shown in this study, the main problem lies in precariousness and the quality of employment (very low wages, temporariness, etc.). Therefore, migration policies should cover two main objectives: fighting against precariousness and opting for a competitive model that is not based on temporary employment and low wages.
  • Regarding the reverse culture shock that migrants experience when they return to Spain, this study highlights the need to provide them psychological support which helps them to manage the feelings and emotions derived from the weakening of relationships and social ties they had in their homeland, and therefore, from the loss of their identities.

These implications and their applicability have been added to the section ‘Conclusions and New Lines of Research’.

Reviewer 2 Report

General Comments:

This is an important topic and timely study focusing on the recent migration patterns of young Spaniards settled in Chile. More specifically this study analyses “how these young people experience their arrival and establishment in said Latin American country and, in the other hand, how the process of returning and readjusting to Spanish society takes place.” This paper addresses a relevant topic for which research is still limited. Thus, this is an important research topic and fills, in part, a major gap in the return migration literature. My overall assessment is that this study would make an important contribution to the studies of return migration in Latin America-Europe, as well as informative reading for the audience of Social Sciences, but only if some major revisions are undertaken by the author(s).   

Specific Comments:

There is need to expand the Introduction (pp. 1-4). Expand on why is this research topic important and how it will fill gaps in the literature? What makes this study so unique within the context of the existing literature/return migration in Europe?

Literature Review/Framework (pp. 4-6): The author(s) show familiarity with the literature. However, I suggest the author(s) also expand/review other European literature on return migration. Also identify the major gaps in the literature.

“Materials and Methods” (pp. 6-9). OK. However, I suggest the author(s) comment on the main limitations of this study (e.g., sampling strategies, small sample, etc).

Results (pp. 9-14): Interesting results. How these results (differ or not) from other Spanish/European studies? How the results from this study can inform government migration policies?

“Conclusions” (pp. 14-15). Expand on the main policy implications of this study and areas for further research.

Author Response

First of all, the authors would like to thank the reviewer for their considerations and constructive criticism, which have undoubtedly helped to improve the work presented here.

Point 1: There is need to expand the Introduction (pp. 1-4). Expand on why is this research topic important and how it will fill gaps in the literature? What makes this study so unique within the context of the existing literature/return migration in Europe?

Response 1: In response to this first suggestion, it has been indicated why the present research is relevant and how it contributes to fill the gap in the existing literature:

“It is worth noting that the present research on recent Spanish migration to Chile is a relevant study of great interest in the field of social sciences. This is so because there is scarce literature on the current migration flows between Europe and Latin America (Sassone and Yépez 2014; Córdova 2015; Pérez‑Caramés et al. 2018; Rodríguez‑Fariñas et al. 2015), and said literature focuses mainly on migration factors and labor market insertion, without dealing in depth, as our study does, with the adaptation processes in the host society and the return process, which makes this research and its results more complex and dynamic, since it contemplates all stages of the migration process—first steps, adaptation process and return.”

That paragraph has been added at the end of the introductory section of the article, along with a new, more recent bibliographical reference:

Rodríguez-Fariñas, M.; Romero-Valiente, J.; Hidalgo-Capitán, A. Los exiliados económicos: La tercera oleada de emigración española a Chile (2008-2014). Revista de geografía Norte Grande. 2015, 61, 107-133

Point 2: Literature Review/Framework (pp. 4-6): The author(s) show familiarity with the literature. However, I suggest the author(s) also expand/review other European literature on return migration. Also identify the major gaps in the literature.

Response 2: As suggested, the literature on return migration has been reviewed and expanded. Thus, the factors on which such literature focuses (both in general and in the Spanish context) have been explained, as well as how our study contributes to expand the knowledge about it. Mainly, it can be seen that these studies analyze the factors that predispose migrants to leave, to return (or not to return) and how labor market insertion occurs after they return. In this case, our research broadens these ideas by adding the importance that migrants attach to perceived social welfare, and above all, by analyzing the reverse cultural shock that migrants experience during the process of readaptation to Spanish society, and how, sometimes, this process affects their identities and subjectivities.

This revision/expansion has been incorporated after Table 1, in section 2: Framework of Analysis.

Likewise, the following references have been added to the bibliography section:

  • Josa‑Prado, F.; Gómez-Sebastián, S. Report on the return and incorporation of scientists and researchers to Spain. CRE: Spain, 2019, pp. 1-44.
  • Arango, J. Spain: new emigration policies needed for an emerging diaspora. Transatlantic Council on Migration and Migration Policy Institute: Washington, D.C., 2016, pp. 1-24.
  • Elgorriaga, I.; Ibabe, I.; Arnoso, A. Intention to return of Spanish emigrant population and associated psychosocial factors. International Journal of Social Psychology. 2020, 2, 413-440.
  • Domínguez‑Mujica, J.; Díaz‑Hernández, R. The Dilemma of Returning: the Liquid Migration of Skilled Spaniards 8 years down the Economic Crisis. Canadian Studies in Population. 2019, 46, 99-119.

Point 3: “Materials and Methods” (pp. 6-9). OK. However, I suggest the author(s) comment on the main limitations of this study (e.g., sampling strategies, small sample, etc).

Response 3: In this case, there were no major limitations to comment on. As we explained in the methodological section, the research was guided by the criterion of theoretical saturation, that is, new interviews were not carried out when they did not provide additional information relevant to the research objectives. Therefore, according to this criterion typical of the Grounded Theory, the number of interviews (sample) is not the most relevant factor, but rather that the interviewees’ discourses help to saturate the sociocultural insertion categories studied (convert, nostalgic and cosmopolitan) and to understand how the readaptation process in Spain took place.

Point 4: Results (pp. 9-14): Interesting results. How these results (differ or not) from other Spanish/European studies? How the results from this study can inform government migration policies?

Response 4: The results differ mainly in that, in addition to analyzing migration factors, they explain how the processes of adaptation to Chilean society and readaptation to Spanish society take place. Therefore, the analysis carried out covers the entire migration process (initiation, adaptation and return), which makes this research and its results more complex and dynamic, since most of the published studies focus on one of these stages (mainly on understanding the factors that led people to emigrate). These differences have been pointed out in the introductory section of the article.

As to how the results can help/inform migration policies, the following paragraph has been added to the conclusions section:

“It is also worth noting the political implications that the results of this study may have. Thus, in words of the interviewees themselves, the main factors that led to the departure of these qualified migrants were the precariousness and temporary nature of the Spanish labor market. Therefore, all government actions and policies carried out to curb these departures should prioritize two objectives: fighting precariousness and opting for a competitive model not based on temporary employment and low salaries. In addition to that, with regard to the reverse cultural shock experienced by these young people after their return, this study highlights the need to provide them with psychological support to manage the feelings and emotions derived from the weakening and/or loss of the identities, relationships and social ties they had in Spain, and which have been greatly transformed during this process.”

Point 5: Conclusions (pp. 14-15). Expand on the main policy implications of this study and areas for further research.

Response 5: The results of this study have implications at three levels:

  • First, they show how part of this emigration responds to a feeling of exile, and therefore, they make visible that one of the main migration factors lies in the precariousness of the Spanish labor market and in the lack of a productive framework capable of incorporating the most qualified labor force.
  • Second, this research shows how the socio‑cultural insertion of these young people in the country of destination takes place, highlighting the importance that migrants give to perceived social welfare (even more so than job stability).
  • And finally, regarding the return process, the discourses of the interviewees show the reverse cultural shock they experience, in which they perceive the loss of their basic social relations and their former spaces of comfort, and therefore, of their identities.

As explained in the response to the previous suggestion, these levels and their political applicability have been pointed out in the conclusions section.

Round 2

Reviewer 2 Report

I recommend this paper  for publication. The author(s) addressed my questions/concerns. However, there is room here to improve the quality of this paper ("Moderate English changes required").